# A New Finding on Magnetic Resonance Imaging for Diagnosis of Hemifacial Spasm with High Accuracy and Interobserver Correlation

**DOI:** 10.3390/brainsci13101434

**Published:** 2023-10-09

**Authors:** Guilherme Finger, Kyle C. Wu, Joshua Vignolles-Jeong, Saniya S. Godil, Ben G. McGahan, Daniel Kreatsoulas, Mohammad T. Shujaat, Luciano M. Prevedello, Daniel M. Prevedello

**Affiliations:** 1Department of Neurosurgery, The Ohio State University College of Medicine, Columbus, OH 43210, USA; guilhermefingermd@gmail.com (G.F.); kyle.wu@osumc.edu (K.C.W.); daniel.kreatsoulas@osumc.edu (D.K.); 2College of Medicine, The Ohio State University College of Medicine, Columbus, OH 43210, USA; joshua.vignolles-jeong@osumc.edu; 3Department of Radiology, The Ohio State University College of Medicine, Columbus, OH 43210, USAluciano.prevedello@osumc.edu (L.M.P.)

**Keywords:** hemifacial spasm, diagnosis, facial nerve disorders, facial nerve disease, magnetic resonance imaging

## Abstract

Among patients with clinical hemifacial spasm (HFS), imaging exams aim to identify the neurovascular conflict (NVC) location. It has been proven that the identification in the preoperative exam increases the rate of surgical success. Despite the description of specific magnetic resonance image (MRI) acquisitions, the site of neurovascular compression is not always visualized. The authors describe a new MRI finding that helps in the diagnosis of HFS, and evaluate the sensitivity, specificity, and interobserver correlation of the described sign. A cross-sectional study including cases of hemifacial spasm treated surgically from 1 August 2011 to 31 July 2021 was performed. The MRIs of the cases were independently evaluated by two experienced neuroradiologists, who were blinded regarding the side of the symptom. The neuroradiologists were assigned to evaluate the MRIs in two separate moments. Primarily, they evaluated whether there was a neurovascular conflict based on the standard technique. Following this initial analysis, the neuroradiologists received a file with the description of the novel sign, named Prevedello Sign (PS). In a second moment, the same neuroradiologists were asked to identify the presence of the PS and, if it was present, to report on which side. A total of 35 patients were included, mostly females (65.7%) with a mean age of 59.02 (+0.48). Since the 35 cases were independently evaluated by two neuroradiologists, a total of 70 reports were included in the analysis. The PS was present in 66 patients (sensitivity of 94.2%, specificity of 91.4% and positive predictive value of 90.9%). When both analyses were performed in parallel (standard plus PS), the sensitivity increased to 99.2%. Based on the findings of this study, the authors conclude that PS is helpful in determining the neurovascular conflict location in patients with HFS. Its presence, combined with the standard evaluation, increases the sensitivity of the MRI to over 99%, without increasing risks of harm to patients or resulting in additional costs.

## 1. Introduction

Hemifacial spasm is characterized by unilateral involuntary spasms of the facial musculature [1] with an annual incidence of 1/100,000 [2] and a prevalence ranging from 9.8 to 11 cases per 100,000 [2,3].

Neurovascular conflict is the main cause of HFS. Identification and evaluation of the NVC via preoperative imaging are critical to guide surgical procedure and technique [4]. The main goal of the preoperative image exam is to identify the point at which the facial nerve is being compressed by a vascular structure. If this NVC is identified preoperatively, surgical treatment has a higher chance of success due to the surgeon’s more comprehensive understanding of the anatomy surrounding the NVC that must be navigated through to release the nerve.

The aim of MRI investigation is to identify the facial nerve and to search for any vascular attachment or compression. The combination of high-resolution 3D T2-weighted imaging with 3D time-of-flight angiography and 3D T1-weighted gadolinium-enhanced sequences is considered the standard for the investigation of primary NVC [5,6,7,8]. However, MRI exploration has limitations. The nature of the conflicting vessel(s) is sometimes difficult to assess, particularly if the area explored by thin sections does not capture the vessel or nerve’s origin. Other reasons described that diminish the efficacy of the MRI are very small caliber of the involved vessel, patients with “small posterior fossa”, or crowded cisternal contents [8]. Therefore, the description of a new image pattern to diagnose the NVC among patients with HFS is valid and helpful to increase the sensibility and accuracy of MRI as the ideal image exam to achieve preoperative diagnoses in these patients.

The authors conducted a study in order to describe a new MRI finding for the diagnosis of HFS and to evaluate its sensitivity, specificity, prevalence ratio, and interobserver identification correlation.

## 2. Materials and Methods

### 2.1. Definition

When the trajectory of the vessel is evaluated in the coronal view, it may be possible to visualize the vessel forming a loop superior and medial (towards the exit of the facial nerve) (Figure 1). When the MRI is visualized in the axial view, at the highest level of the loop in the coronal section, the proximity of the vascular structures to the pons may be seen in the axial view as if the artery was located inside of the brainstem, surrounded by the pons parenchyma (Figure 2). 

The Prevedello Sign is the identification of an arterial loop enfolded by the pons near the exit of the facial nerve, in the brainstem. This finding is best visualized in T1WI with contrast and in FIESTA acquisitions of MRI sequences (Figure 3).

### 2.2. Study Design

The study was conducted at the Department of Neurosurgery, Skull Base section, at the Ohio State University and approved by the Ethics Review Board under the number 2022E0740.

The authors conducted a cross-sectional study evaluating all cases of hemifacial spasm treated surgically from 1 August 2011 to 31 July 2021. A retrospective chart review of the surgical records of the hospital’s database was performed in order to identify all patients diagnosed with hemifacial spasm and had been treated surgically. 

Two experienced neuroradiologists evaluated all cases independently. They were aware that the patients were clinically diagnosed with HFS, but they were blinded from knowing the symptomatic side. They evaluated each case two different times. Primarily, they were asked to analyze the MRI in order to evaluate whether there was a neurovascular conflict based on the gold standard technique which consists of identifying the facial nerve and accurately evaluating its entire path from the pontomedullary transition to the internal acoustic canal and to search for any vascular attachment or compression. Once they had finished this initial analysis, the neuroradiologists received a file with the description of the PS. In a second instance, the same neuroradiologists were asked to reassess the same MRI list, looking to identify the presence of this novel sign and to determine on which side it was present. 

Once the authors received the reports from both neuroradiologists, we analyzed the accuracy of the diagnosis based on the standard evaluation (percentage of diagnosis in the symptomatic sign) and also the presence of the PS and its accuracy (to calculate sensibility). The authors utilized the contralateral side (asymptomatic side) as the control to calculate the specificity. 

### 2.3. Image Study

The MRI examinations were performed on a 3 T MR scanner. The MRIs were performed using our cranial nerve protocol, which were sagittal, and axial T2-heavily weighted images (T2WI) in thin-sliced acquisitions; axial, sagittal, and coronal T1 weighted images (T1WI) with contrast of the whole brain and also a thin-slice sequence of the posterior fossa; and axial fluid-attenuated inversion recovery (FLAIR) and diffusion-weighted imaging (DWI) sequences.

The image sequences were obtained in 0.3 to 0.4 mm slice thickness and a multiplanar oblique reconstructions were obtained in axial, sagittal, and coronal views, following the course of the facial nerve.

### 2.4. Variables

The variables analyzed included radiological, clinical, and demographic-related variables. The radiological related variables included the presence of the NVC according to the standard criteria established in the medical literature, the presence or absence of the PS, and the side on which it was identified. The clinical variables analyzed included the side of the symptom and the vessel responsible for the NVC. The possible arteries involved in the NVC were the vertebral artery (VA), posterior inferior cerebellar artery (PICA), and anterior inferior cerebellar artery (AICA). When the compression involved more than one artery, all of the arteries were described as contributing to the NVC (i.e., AICA and VA, PICA, and AICA). The demographic data evaluated was the age and gender of the patients.

### 2.5. Patient Eligibility

Patients admitted at The Ohio State University Wexner Medical Center between 1 August 2011 and 31 July 2021, who were diagnosed with HFS and submitted to microsurgical neurovascular decompression were included. 

### 2.6. Statistical Analysis

Data were collected using Microsoft Excel 2019 software. Statistical analysis was performed using the Statistical Package for the Social Sciences (IBM SPSS Statistics for Windows, Version 22.0. Armonk, NY, USA: IBM Corp.). Numerical variables parametricity were tested and parametric distributed variables were presented as mean and standard deviation. Categorical variables were presented in absolute numbers and proportion. The sensitivity, specificity, and Cohen’s Kappa Index were calculated for the analyses based on the results of the standard evaluation and the presence of the PS. 

The prevalence ratio of the PS was calculated according to the formula: percentage of PS in the symptomatic sign divided by the percentage of PS in the asymptomatic sign.

Parallel combined sensitivity was calculated using the following formula: sensitivity of standard method + sensitivity of PS − [sensitivity of standard method × sensitivity of PS].

Parallel combined specificity was calculated according to the formula: specificity of standard method × specificity of PS. 

## 3. Results

During the study period, a total of 35 patients were surgically treated for HFS. The majority of patients were females (65.7%) with a mean age of 59.02 (+0.48). According to the symptoms, the right side was involved in 13 cases (37.14%) and the left side in 22 cases (62.85%). There were no cases of bilateral hemifacial spasm in this sample (Table 1).

The intraoperative vascular compression was identified in all cases, but in one case the vessel was not specified. The PICA was solely responsible for the NVC in 11 cases (31.42%), combined PICA and AICA arteries were responsible for NVC in 6 cases (17.14%), solely AICA in 8 cases (22.85%), combined PICA and VA in 5 cases (14.28%), and the VA alone was responsible for 4 cases (11.42%) of NVC in this sample.

Since two different neuroradiologists evaluated the 35 cases searching for the presence of neurovascular compression on both sides, a total of 140 evaluations were performed.

Since all patients presented symptoms unilaterally and both neuroradiologists independently evaluated each case but they were blinded from the symptomatic side, they were forced to evaluate both sides in every patient; this totaled 140 reports that were included in the analysis (35 cases × 2 sides evaluated × 2 neuroradiologists). 

Based on MRI analysis using the standard technique, 66 neurovascular compressions were identified on the symptomatic side (sensitivity of 94.2%, specificity of 97.1%, PPV of 97.05% and NPV of 94.4%). The PS was present in 66 analyses (sensitivity of 94.2%) and among the 70 analyses of the asymptomatic side, 64 did not have the PS (specificity 91.4%). Among the patients in whom the PS was present, 90.9% of the time it was identified on the symptomatic side (PPV 90.9%) (Table 2). The calculated prevalence ratio was 1.41.

When both analyses were performed in parallel (standard plus PV), the sensitivity increased to 99.2%.

The correlation Cohen Kappa index was performed for both analyses, demonstrating an index of 0.714 for the gold standard evaluation and 0.542 for the Prevedello sign.

## 4. Discussion

Primary HFS is triggered by neurovascular conflict (NVC), whereas secondary HFS comprises all other causes that may compress the facial nerve, such as posterior fossa tumors, arteriovenous malformation, Paget’s disease, and Chiari malformation [7]. 

The facial nerve exits the brain stem as a single entity forming the cisternal segment extending towards the internal acoustic meatus with a mean length of 17.93 mm (range, 14.8–20.9 mm) [9]. Along its path, the facial nerve runs close to vascular structures. If these structures are in contact with the nerve, an NVC may occur. According to the literature, the AICA is the most common vessel causing NVC (corresponding for 43% of the cases), followed by the PICA (31%) and VA (23%) [3,10]. In our series, the PICA was the most frequently involved artery solely responsible for NVC and was involved in additional instances in conjunction with the AICA or the VA. The AICA alone and the VA alone were responsible for one third of the cases.

In the majority of cases, the location of the neurovascular compression is in the first few millimeters from the brainstem. The facial nerve emerges in the brainstem surface from the pontomedullary sulcus at the upper edge of the supraolivary fossette and strongly adheres to the surface of the pons for 8–10 mm before separating from the brainstem [10]. Once separated from the brainstem, the first 1.9 to 2.86mm of the facial nerve is a segment characterized by the transition of cells responsible for myelination of the nerve (from oligodendrocytes to the Schwann cells) [3,11], and is anatomically known as root exit/entry zone of cranial nerve [12]. Histologically, this specific area of the nerve is defined as Obersteiner–Redlich transition zone (TZ). Since this zone lacks an epineurium, the nerve is protected by an arachnoid membrane only, which makes it vulnerable to mechanical compression caused by vessel’s pulsation [13]. Pressure applied in this region may trigger action potentials from the demyelinated segment of the nerve, leading to symptoms [14]. 

Regular MRI sequences are essential in determining secondary causes of HFS [7,15]. Post-contrast T1 sequences (to evaluate cerebellopontine angle solid tumors), T2-weighted sequence (to evaluate intra-axial lesions in the brainstem, i.e.: demyelinating disease), and diffusion-weighted imaging (to evaluate for the presence of cystic disease, i.e., differentiating epidermoid cysts from arachnoid) are useful for the differential diagnosis.

In the context of primary NVC, since it mainly occurs at the REZ, this segment should be carefully explored [8]. Image slice thickness should optimally be between 0.3 to 0.4mm [7,16], especially in the arterial phase [17,18]. 

Multiplanar oblique reconstructions should be obtained following the course of the facial nerve. A variety of high-resolution 3D heavily T2WI sequences are currently available, depending on the manufacturer, and may be helpful during investigation. These sequences include constructive interference in steady-state (CISS), steady-state-free precession (SSFP), T2WI-driven equilibrium radio frequency reset pulse (DRIVE), three-dimensional fast imaging employing steady-state acquisition (FIESTA), and sampling perfection with application-optimized contrasts by using different flip angle evolutions (SPACE) [7,19]. Additionally, 3D T2WI sequences provide “cisterno-graphic” images of cranial nerves and vessels surrounded by cerebrospinal fluid, with high spatial resolution [7,16].

Despite the various imaging modalities described above, in some patients the NVC is not visualized. According to this sample analysis, 6 patients out of a 100 will not have the diagnosis of the NVC using the standard MRI method. This scenario puts the neurosurgeon in a challenging situation, since not indicating surgical intervention will prolong the patient’s symptomatology and increase their morbidity when it could be resolved surgically (this would represent an error type 1/alpha, due to the limitation of the diagnosis method). On the other hand, indicating surgery without preoperative visualization of an NVC on image exams increases the risk of surgery misindication or surgical failure.

An interesting paper published by Ahmad et al. [20], investigated the relationship between practice setting of radiologist interpreting MRI scans and reported detection of NVC in patients with trigeminal neuralgia (TN), whose neurovascular compression was confirmed intraoperatively. According to their results, blinded academic neuroradiologists are more likely to detect neurovascular compression when compared with community radiologists. Although this paper included only TN cases, the physiopathology, investigative methods, and treatment of TN are similar to HFS; therefore, we can assume that the variability reported in papers related to TN may also occur in the evaluation of HFS [21,22].

The advent of a second independent method to determine the NVC is helpful to increase the diagnosis rate. Even though the PS used independently had a lower sensitivity and specificity when compared to the standard criteria, the use of both diagnostic methods in conjunction increased the sensitivity to 99%.

This new MRI criteria is helpful for the diagnosis of NVC in patients with HFS, not in terms of being a substitute of the standard pattern, but as a complementary diagnostic method.

This paper has several limitations that should be highlighted. The sample size of this study was determined by the number of patients meeting the inclusion criteria rather than based on sample size calculation. The sample size of 35 patients may influence the sensitivity of the MRI evaluation, and a higher number of patients would better approximate the epidemiologic values calculated in this study. A single center study may bias the evaluation of the MRI and also the agreement between the neuroradiologists (even though they evaluated the cases independently, the proximity and weekly radiological discussion of skull base cases may interfere in the evaluation). The patients were not submitted to a 3D time-of-flight magnetic resonance angiography, which was recently described as having the better diagnostic performance for detecting NVC in patients with trigeminal neuralgia or HSF [23]. On the other hand, the interpretation of the exams by neuroradiologists focused on skull base pathologies may overestimate the sensitivity of the standard MRI evaluation. The moderate correlation index for the PS may be due to the low familiarity of the neuroradiologists with this sign, but also may represent variable interpretation of the sign. Finally, the purpose of the study was to describe a new MRI finding for HFS and to analyze its frequency patterns. The MRI images were retrospectively assessed; therefore, the presence of the PS was not used as a criterion for surgery indication in the cases included in this sample. However, it is important to emphasize that the senior author has been using this sign as an easy method of corroboration with the patient’s symptomatology, which is the most important factor to indicate surgery. In addition, the study’s design (cross-section) is not ideal for evaluating outcomes. To determine if the presence of the PS helps to achieve a better surgical result, it is necessary to perform a longitudinal study (either prospective or retrospective, which are cohort and case-control, respectively).

Despite the limitations discussed above, the main strength of this paper is to describe an MRI sign that is helpful in the diagnosis of NVC for HFS in cases where the standard diagnosis is doubtful or not visualized. Additionally, the evaluation of the PS is performed with regular MRI protocols and does not require any additional or different image acquisitions (it does not increase the cost of the exam, extra gadolinium infusion is not necessary, and it does not present any increased risk for the patients). The only additional work required to evaluate the presence of the PS is a supplementary examination of the T1WI and FIESTA axial acquisitions by the radiologists to evaluate for the presence of the PS.

## 5. Conclusions

The PS is helpful in determining the neurovascular conflict point for diagnosis in patients with HFS. The presence of this sign, combined with the standard evaluation, increases the sensitivity of the MRI to over 99%, without increasing risk of harm to patients or resulting in additional costs.

## Figures and Tables

**Figure 1 brainsci-13-01434-f001:**
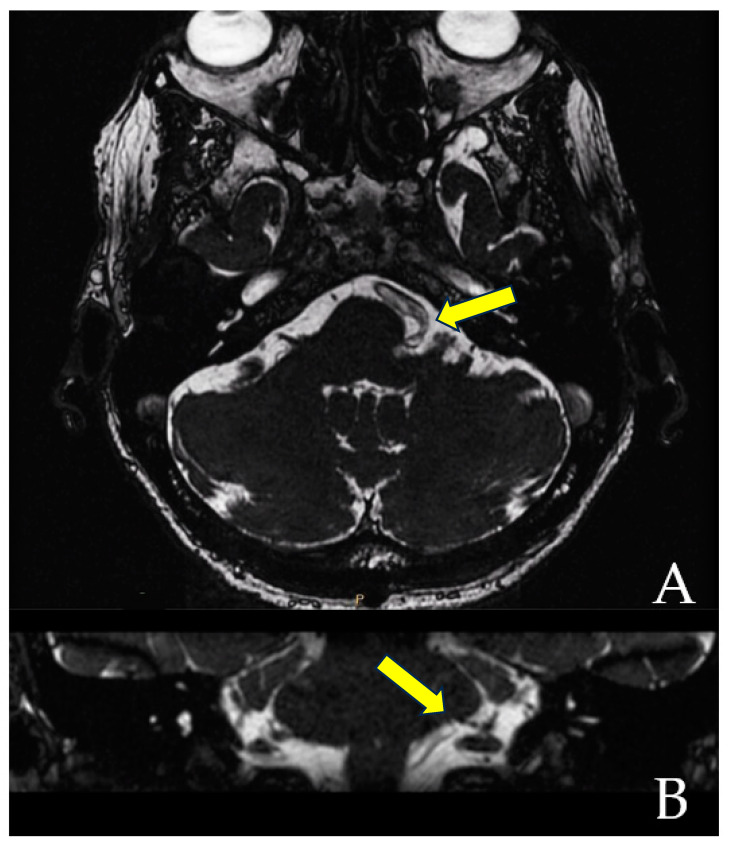
The axial view gives the false impression that the artery is located inside the pons (**A**). However, the coronal view clearly demonstrates that the vessel does not enter the brainstem (**B**). The yellow arrows point to the local of the NVC.

**Figure 2 brainsci-13-01434-f002:**
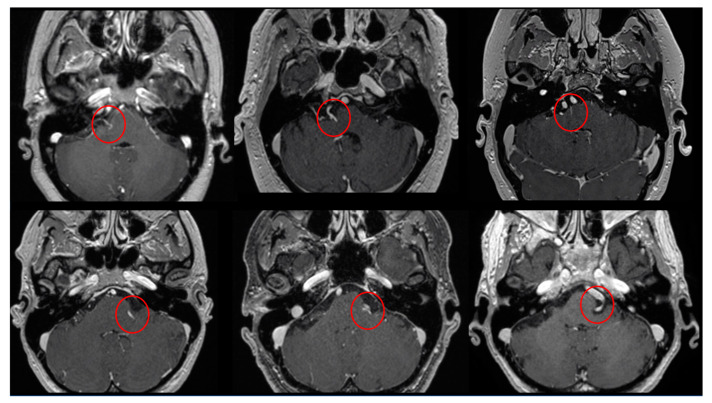
Six examples of the PS identified (inside the red circles) in T1WI with gadolinium axial images.

**Figure 3 brainsci-13-01434-f003:**
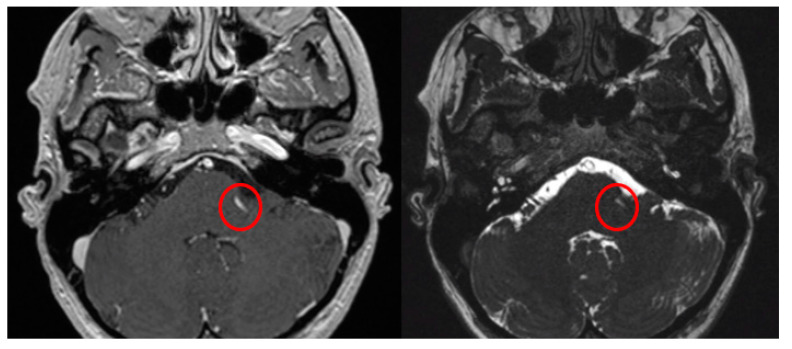
Axial view of MRI in T1WI with gadolinium and FIESTA sequences demonstrating the presence of the Prevedello sign (an arterial loop attached to the pons near the exit of the facial nerve), highlighted by the red circles.

**Table 1 brainsci-13-01434-t001:** Epidemiological and clinical results.

Age	59.02 (±0.48) *
Gender	
Female	23 (65.7%) °
Male	12 (34.2%) °
Symptomatic side	
Right	13 (37.14%) °
Left	22 (62.85%) °
Artery involved	
PICA	11 (31.42%) °
AICA	8 (22.85%) °
VA	4 (11.42%) °
AICA + PICA	6 (17.14%) °
PICA + VA	5 (14.28%) °

* Mean (±standard deviation), ° Absolute number (percentage).

**Table 2 brainsci-13-01434-t002:** Test analysis results.

	Sensitivity	Specificity	PPV	NPV	Cohen Kappa Index
Standard MRI evaluation	94.2%	97.1%	97.05%	94.4%	0.714
Prevedello sign	94.2%	91.4%	90.9%		0.82
Tests in parallel	99.2%	88.7%			

PPV: positive predictive value, NPV: negative predictive value.

## Data Availability

The data for this study is unavailable due to privacy or ethical restrictions in concordance to the institutional HIPAA’s policy.

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
