# Peer review of "A New Finding on Magnetic Resonance Imaging for Diagnosis of Hemifacial Spasm with High Accuracy and Interobserver Correlation"

_brainsci, 2023, doi:10.3390/brainsci13101434_

Round 1

Reviewer 1 Report

1. What is the main question addressed by the research?
- A New Finding on Magnetic Resonance Imaging for Diagnosis of Hemifacial Spasm

2. Do you consider the topic original or relevant in the field? Does it
address a specific gap in the field?
- Yes, it is original and relevant to be discussed.

3. What does it add to the subject area compared with other published
material?
- It adds a new sign. However, the authors should define the proposed new imaging finding.

4. What specific improvements should the authors consider regarding the methodology? What further controls should be considered?
- The authors should describe the power of the study. Also, the numbers do not correspond. They should better describe the results.

5. Are the conclusions consistent with the evidence and arguments presented, and do they address the main question posed?
- Yes

6. Are the references appropriate?
- Yes

7. Please include any additional comments on the tables and figures.
- The authors should describe that their paper was already presented in part at a conference. 

Please also solve the following issues.

1.     Please revise the abstract. The number described in the abstract is not corresponding to the total number of individuals studied.

2.     How was the power of the study calculated?

3.     Please define what the authors mean by the Cobra Cave and Prevedello signs. The text definition is difficult to understand.

4.     Please describe why this is the name Cobra Cave sign.

5. State that this manuscript was already presented as an abstract.

Author Response

  1. Please revise the abstract. The number described in the abstract is not corresponding to the total number of individuals studied.

Answer: The authors appreciate the reviewer suggestion. In fact the numbers correspond, since 35 cases were evaluated by 2 different neuroradiologists, a total of 70 evaluations were included. The authors agree with the reviewer that this information was not clear in the abstract and may be a source of confusion. The authors rewrote the abstract and included the following statement to be clearer for the readers: “Since the 35 cases were independently evaluated by two neuroradiologists, a total of 70 reports were included in the analysis.”

  1. How was the power of the study calculated?

Answer: The authors appreciate the reviewer’s suggestion. However, the authors performed a cross-sectional study which scope was to describe a new MRI finding for the diagnosis of hemifacial spasm. The authors were not testing a hypothesis and the study design was not longitudinal. For these reasons the power of the study (1- type 2 error) was not calculated. There are modules that might estimate the power of cross-section studies. However, the authors reinforce that the no hypothesis were being tested, so we believe the calculation of the power is not suitable.

Based on the reviewer suggestion, the authors calculated the prevalence ratio of the PS in patients with HFS and found a value of 1.41. We added this information to the results section.

  1. Please define what the authors mean by the Cobra Cave and Prevedello signs. The text definition is difficult to understand.

Answer: The authors decided to change the name Cobra Cave Sign to Prevedello Sign, which is how it is referred in our institution. Besides, we added more information in the definition section to better explain the sign as follows:

“When the trajectory of the vessel is evaluated in the coronal view, it may be possible to visualize the vessel forming a loop superior and medial (towards the exit of the facial nerve) (Figure 1). When the MRI is visualized in the axial view, at the highest level of the loop in the coronal section, the proximity of the vascular structures to the pons may be seen in the axial view as if the artery was located inside of the brainstem, surrounded by the pons parenchyma (Figure 2). The Prevedello Sign is the identification of an arterial loop enfolded by the pons near the exit of the facial nerve, in the brainstem. This finding is best visualized in T1WI with contrast and in FIESTA acquisitions of MRI sequences (Figure 3).”

  1. Please describe why this is the name Cobra Cave sign.

Answer: The authors decided to change the name Cobra Cave Sign to Prevedello Sign, which is how it is referred in our institution.

  1. State that this manuscript was already presented as an abstract.

Answer: The authors included the following paragraph to the manuscript: This study was presented as oral presentation at the North American Skull Base Society 32nd Annual Meeting in Tampa/FL, 2023. This statement can be found as a Previous presentation section following the conflict of interest section.

Reviewer 2 Report

The  study is interesting but deserves some extensive revision from a methodological point of view. For instance, the authors should specify, for each patient, if a neurovascular conflict was detected or not and, if not, if CCS could help in such cases. Then they should confront it with the "classical" appearance of neurovascular conflict and prove it is better at detecting conflicts. Otherwise, proposing a new radiological sign is pointless.

English reviewing is encouraged. Typos are present (see the first lines of the abstract).

Author Response

The  study is interesting but deserves some extensive revision from a methodological point of view. For instance, the authors should specify, for each patient, if a neurovascular conflict was detected or not and, if not, if CCS could help in such cases. Then they should confront it with the "classical" appearance of neurovascular conflict and prove it is better at detecting conflicts. Otherwise, proposing a new radiological sign is pointless.

Answer: The authors appreciate the reviewer suggestion and respect the point of view described. The authors described in the results section the percentage of patients who had the neurovascular conflict identified based the classic evaluation (sensitivity of 94.2% and specificity of 97.1%), and also the results when the Prevedello sign was evaluated (sensitivity of 94.2% and specificity 91.4%).  The Prevedello sign alone did not show a better accuracy when compared to the classic evaluation. However, at our perspective, it does not mean that proposing this new radiological sign is pointless, since when the search for the Prevedello sign is added to classic evaluation, the increases the sensitivity of the MRI to over 99%. And, as described in the manuscript, the search for this sign only requires 1 more minute of extra evaluation of the MRI, there is no need to order a different exam, nor different sequences that are usually performed in the investigation of the Hemifacial spasm. In our perspective, this is a 1 extra minute evaluation that allows a 99% sensitivity of the MRI; which seems valid (at least for us). Once again, we reinforce that the search does not increase risk of harm to patients or resulting in additional costs.

This concept is analogous to the diagnosis of diabetes and Cushing`s Disease.

English reviewing is encouraged. Typos are present (see the first lines of the abstract)

Answer: The authors are very thankful for the reviewer suggestion and inform that the manuscript was reviewed.

Reviewer 3 Report

The article and the sign described are very interesting and useful, but there are some questions and corrections:

1.       Figure 1 is not representative at all. The aim of the article is to introduce the new MRI sign, so you should give more examples of fine quality (one Figure includes several cases – zoom on the CCS). It is highly recommended to include a scheme or a picture of the described sign as well. Why is it named cobra cave?

2.       Figure 2: Arrows are needed, it's difficult to understand.

3.       Did the patients included into the study improve after surgery? Did all of them have the conflict in the CCS area?

4.       MRI protocol should be described in detail, especially the regimens crucial for conflict detection. It's not written here about T2-heavily weighted images or angiography of contrast enhancement and their characteristics (slice thickness, TR, TE, FOV, etc.), although authors postulate these regimens to be standard in this pathology.

5.       It's a mess about the number of patients – first, authors wrote 35 patients, then “ 66 identified on the symptomatic side (line 141)” – but there were only 35 patients, weren’t there? And then (lines 142-144): “The CCS was present in 66 patients (sensitivity of 85.7%) and among the 70 analyses of the asymptomatic side, 64 did not have the CCS (specificity 91.4%).” 66 patients??

6.       The sign itself is named as “cobra cave sign”, but then “Prevedello sign” appears—all the names should be the same or introduced in one place during the first mention in the article text.

7.       Almost all the references used are quite old (more than 5 years old), except for a couple of articles.

Author Response

  1. Figure 1 is not representative at all. The aim of the article is to introduce the new MRI sign, so you should give more examples of fine quality (one Figure includes several cases – zoom on the CCS). It is highly recommended to include a scheme or a picture of the described sign as well. Why is it named cobra cave?

Answer: The authors decided to change the name Cobra Cave Sign to Prevedello Sign, which is how it is referred in our institution. We also added MRI images of several cases to better exemplify the MRI pattern of the Prevedello Sign (Figure 2).

  1. Figure 2: Arrows are needed, it's difficult to understand.

Answer: The authors included arrows to better identify the location of the neurovascular conflict as suggested by the reviewer.

  1. Did the patients included into the study improve after surgery? Did all of them have the conflict in the CCS area?

Answer: The authors appreciate the reviewer suggestions. We believe that the answer of both questions were already present at the manuscript and follows. In the paragraph of the limitations of the study we report: “. Besides, the study's design (cross-section) is not ideal for evaluating outcomes. To determine if the presence of the PS helps to achieve a better surgical result, it is necessary to perform a longitudinal study (either prospective or retrospective, which are cohort and case-control, respectively).” In the results section, we describe the following sentence in the second paragraph: “The intraoperative vascular compression was identified in all cases”.   

  1. MRI protocol should be described in detail, especially the regimens crucial for conflict detection. It's not written here about T2-heavily weighted images or angiography of contrast enhancement and their characteristics (slice thickness, TR, TE, FOV, etc.), although authors postulate these regimens to be standard in this pathology.

Answer: The authors appreciate the reviewer suggestion. In the methods section the authors wrote a subheading named “Image study”, which contains the following information:

The MR examinations were performed on a 3 T MR scanner. The MRIs were performed using our cranial nerve protocol, which of sagittal and axial T2-heavily Weighted Images (T2WI) in thin-sliced acquisitions; axial, sagittal and coronal T1 Weighted Images (T1WI) with contrast of the whole brain and also a thin-slice sequence of the posterior fossa; axial Fluid-Attenuated Inversion Recovery (FLAIR) and Diffusion Weighted Imaging (DWI) sequences.

The image sequences were obtained in 0.3 to 0.4 mm slice thickness and a multiplanar oblique reconstructions were obtained in axial, sagittal and coronal views, following the course of the facial nerve.

  1. It's a mess about the number of patients – first, authors wrote 35 patients, then “ 66 identified on the symptomatic side (line 141)” – but there were only 35 patients, weren’t there? And then (lines 142-144): “The CCS was present in 66 patients (sensitivity of 85.7%) and among the 70 analyses of the asymptomatic side, 64 did not have the CCS (specificity 91.4%).” 66 patients??

The authors are thankful for the reviewer suggestion. We understand that it may be a little confusing but, as described in the methodology if 35 patients were included and independently evaluated by 2 neuroradiologists, we have a total of 70 evaluations, as expected. Since all patients had a clinical diagnosis of unilateral hemifacial spasm, we expect to have 70 side with conflict and 70 sides without conflict. When analyzing the 70 sides with conflict, the PS was present in 66 (sensitivity of 94.2%). When analyzing the 70 sides without conflict/asymptomatic, the PS was not present in 64 (specificity of 91.4%).

  1. The sign itself is named as “cobra cave sign”, but then “Prevedello sign” appears—all the names should be the same or introduced in one place during the first mention in the article text.

Answer: The authors decided to change the name Cobra Cave Sign to Prevedello Sign, which is how it is referred in our institution.

  1. Almost all the references used are quite old (more than 5 years old), except for a couple of articles.

Answer: The authors appreciate the reviewer suggestion. We performed a new search in Pubmed with the keywords: hemifacial spasm AND diagnosis AND/OR magnetic resonance/MRI. We found one paper published in 2023 that was included in the reference list:

Liang C, Yang L, Reichardt W, Zhang B, Li R. Different MRI-based methods for the diagnosis of neurovascular compression in trigeminal neuralgia or hemifacial spasm: A network meta-analysis. J Clin Neurosci. 2023;108:19-24. doi:10.1016/j.jocn.2022.12.016

Round 2

Reviewer 3 Report

The article was improved greatly after review.